# $\pi^3$: Permutation-Equivariant Visual Geometry Learning

**Yifan Wang**[1,2*]   **Jianjun Zhou**[2,3,4*]   **Haoyi Zhu**[2,5]   **Wenzheng Chang**[1,2]   **Yang Zhou**[2,6]
**Zizun Li**[2,5]   **Junyi Chen**[1,2]   **Jiangmiao Pang**[2]   **Chunhua Shen**[4]   **Tong He**[2,3†]

[1]Shanghai Jiao Tong University   [2]Shanghai AI Laboratory   [3]Shanghai Innovation Institute
[4]Zhejiang University   [5]University of Science and Technology of China   [6]Fudan University
https://github.com/yyfz/Pi3

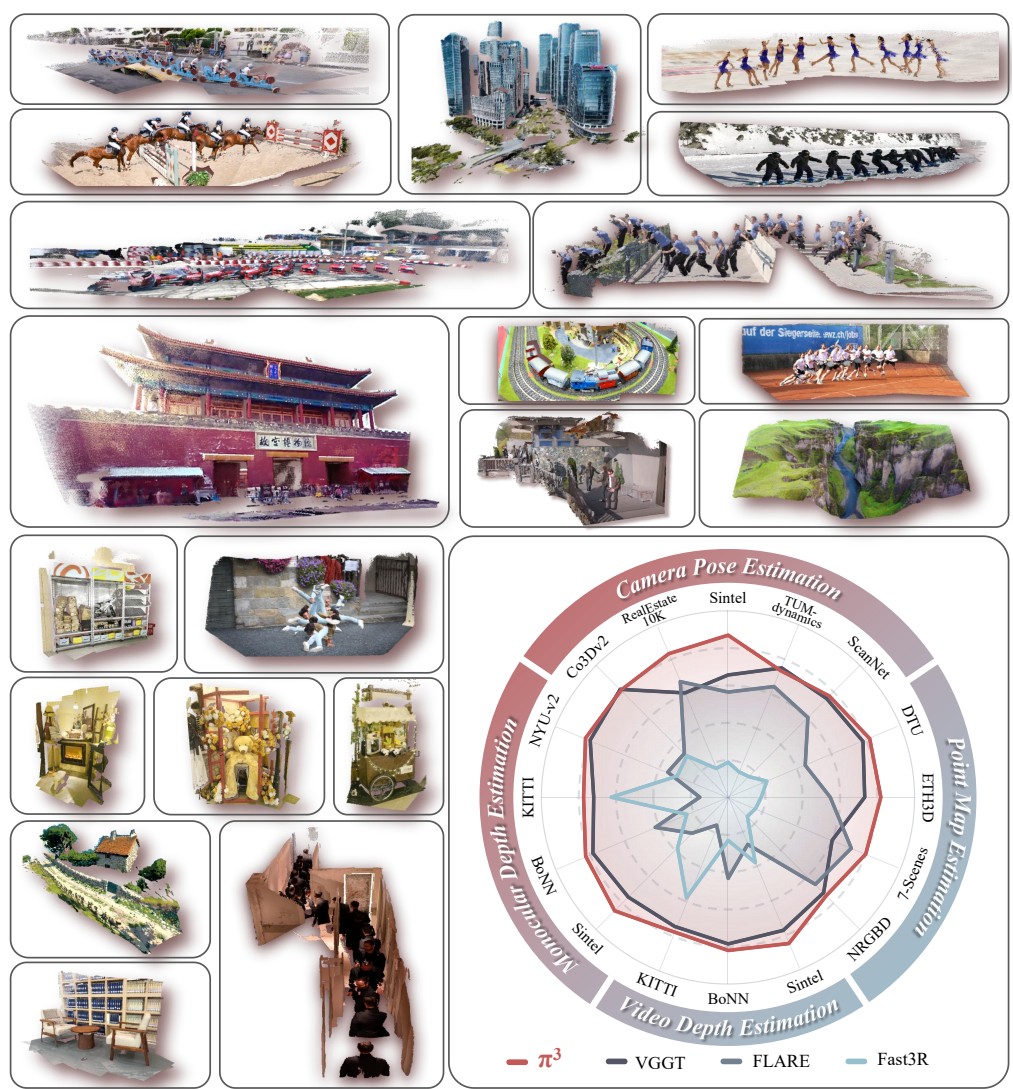

Figure 1: $\pi^3$ effectively reconstructs a diverse set of open-domain images in a feed-forward manner, encompassing various scenes such as indoor, outdoor, and aerial-view, as well as cartoons, with both dynamic and static content.

*Equal Contribution.
†Corresponding Author. Email: tonghe90@gmail.com

## ABSTRACT

We introduce $\pi^3$, a feed-forward neural network that offers a novel approach to visual geometry reconstruction, breaking the reliance on a conventional fixed reference view. Previous methods often anchor their reconstructions to a designated viewpoint, an inductive bias that can lead to instability and failures if the reference is suboptimal. In contrast, $\pi^3$ employs a fully permutation-equivariant architecture to predict affine-invariant camera poses and scale-invariant local point maps without any reference frames. This design not only makes our model inherently robust to input ordering, but also leads to higher accuracy and performance. These advantages enable our simple and bias-free approach to achieve state-of-the-art performance on a wide range of tasks, including camera pose estimation, monocular/video depth estimation, and dense point map reconstruction. Code and models are available at Pi3.

## 1 INTRODUCTION

Visual geometry reconstruction, a long-standing and fundamental problem in computer vision, holds substantial potential for applications such as augmented reality (Engel et al., 2023), robotics (Zhu et al., 2024), and autonomous navigation (Mur-Artal et al., 2015). While traditional methods addressed this challenge using iterative optimization techniques like Bundle Adjustment (BA) (Hartley & Zisserman, 2003), the field has recently seen remarkable progress with feed-forward neural networks. End-to-end models like DUSt3R (Wang et al., 2024) and its successors have demonstrated the power of deep learning for reconstructing geometry from image pairs (Leroy et al., 2024; Zhang et al., 2024), videos, or multi-view collections (Yang et al., 2025; Zhang et al., 2025; Wang et al., 2025a).

Despite these advances, a critical limitation persists in both classical and modern approaches: the reliance on selecting a single, fixed reference view. The camera coordinate system of this chosen view is treated as the global frame of reference, a practice inherited from traditional Structure-from-Motion (SfM) (Hartley & Zisserman, 2003; Cui et al., 2017; Schonberger & Frahm, 2016; Pan et al., 2024) or Multi-view Stereo (MVS) (Furukawa et al., 2015; Schönberger et al., 2016). We contend that this design choice introduces an *unnecessary* inductive bias that fundamentally constrains the performance and robustness of feed-forward neural networks. As we demonstrate empirically, this reliance on an arbitrary reference makes existing methods, including the state-of-the-art (SOTA) VGGT (Wang et al., 2025a), highly sensitive to the initial view selection. A poor choice can lead to a dramatic degradation in reconstruction quality, hindering the development of robust systems (Figure 2).

To overcome this limitation, we introduce $\pi^3$ (Figure 1), a robust, accurate, and fully permutation-equivariant method that eliminates reference view-based biases in visual geometry learning. $\pi^3$ accepts varied inputs—including single images, video sequences, or unordered image sets from static or dynamic scenes—without designating a reference view. Instead, our model predicts an affine-invariant camera pose and a scale-invariant local pointmap, with the pointmap being defined in that frame's own camera coordinate system. By eschewing order-dependent components like frame index positional embeddings and employing a transformer architecture that alternates between view-wise and global self-attention (similar to (Wang et al., 2025a)), $\pi^3$ achieves true permutation equivariance. This guarantees a consistent one-to-one mapping between visual inputs and the re-

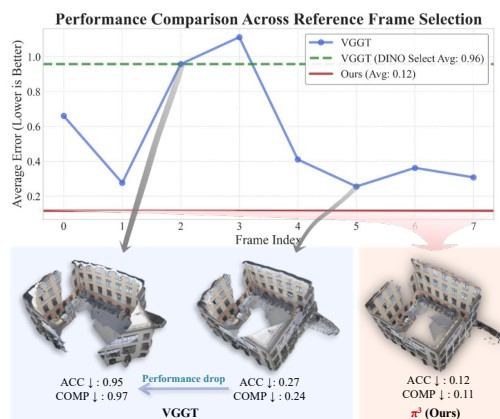

Figure 2: **Performance comparison across different reference frames.** While previous methods, even with DINO-based selection, show inconsistent results, $\pi^3$ consistently delivers superior and stable performance, demonstrating its robustness.

constructed geometry, making the model inher-
ently robust to input order and immune to the reference view selection problem (Table 7).

Our design yields significant advantages. Primarily, it is substantially more robust. Unlike previous methods, our approach demonstrates minimal performance degradation and a low standard deviation when the reference frame is altered (Figure 2 and Table 4.4). Furthermore, it enhances reconstruction accuracy over earlier methods that rely on a reference view.

Through extensive experiments, $\pi^3$ establishes a new SOTA across numerous benchmarks and tasks. For example, it achieves comparable performance to existing methods like MoGe (Wang et al., 2025c) in monocular depth estimation, and outperforms VGGT (Wang et al., 2025a) in video depth estimation and camera pose estimation. On the Sintel benchmark, $\pi^3$ reduces the camera pose estimation ATE from VGGT's 0.167 down to 0.074 and improves the scale-aligned video depth absolute relative error from 0.299 to 0.233. Furthermore, $\pi^3$ is both lightweight and fast, achieving an inference speed of 57.4 FPS compared to DUSt3R's 1.25 FPS and VGGT's 43.2 FPS. Its ability to reconstruct both static and dynamic scenes makes it a robust and optimal solution for real-world applications.

In summary, the contributions of this work are as follows:

- We are the first to systematically identify and challenge the reliance on a fixed reference view in visual geometry reconstruction, demonstrating how this common design choice introduces a detrimental inductive bias that limits model robustness and performance.

- We propose $\pi^3$, a novel, fully permutation-equivariant architecture that eliminates this bias. Our model predicts affine-invariant camera poses and scale-invariant pointmaps in a purely relative, per-view manner, completely removing the need for a global coordinate system.

- We demonstrate through extensive experiments that $\pi^3$ establishes a new state-of-the-art on a wide range of benchmarks for camera pose estimation, monocular/video depth estimation, and pointmap reconstruction, outperforming prior leading methods.

## 2 RELATED WORK

### 2.1 TRADITIONAL 3D RECONSTRUCTION

Reconstructing 3D scenes from images is a foundational problem in computer vision. Classical methods, such as Structure-from-Motion (SfM) (Hartley & Zisserman, 2003; Cui et al., 2017; Schonberger & Frahm, 2016; Pan et al., 2024) and Multi-View Stereo (MVS) (Furukawa et al., 2015; Schönberger et al., 2016), have achieved considerable success. These techniques leverage the principles of multi-view geometry to establish feature correspondences across images, from which they estimate camera poses and generate dense 3D point clouds. Although robust, particularly in controlled environments, these methods typically rely on complex, multi-stage pipelines. Moreover, they often involve time-consuming iterative optimization problems, such as Bundle Adjustment (BA), to jointly refine the 3D structure and camera poses.

### 2.2 FEED-FORWARD 3D RECONSTRUCTION

Recently, feed-forward models have emerged as a powerful alternative, capable of directly regressing the 3D structure of a scene from a set of images in a single pass. Pioneering efforts in this domain, such as Dust3R (Wang et al., 2024), focused on processing image pairs to predict a point cloud within the coordinate system of the first camera. While effective for two views, scaling this to larger scenes requires a subsequent global alignment step, a process that can be both time-consuming and prone to instability.

Subsequent work has focused on overcoming this limitation. Fast3R (Yang et al., 2025) represents a significant advance by enabling simultaneous inference on thousands of images, thereby eliminating the need for a costly and fragile global alignment stage. Other approaches have explored simplifying the learning problem itself. For instance, FLARE (Zhang et al., 2025) decomposes the task by first predicting camera poses and then estimating the scene geometry. VGGT (Wang et al., 2025a) leverages multi-task learning and large-scale datasets to achieve superior accuracy and performance.

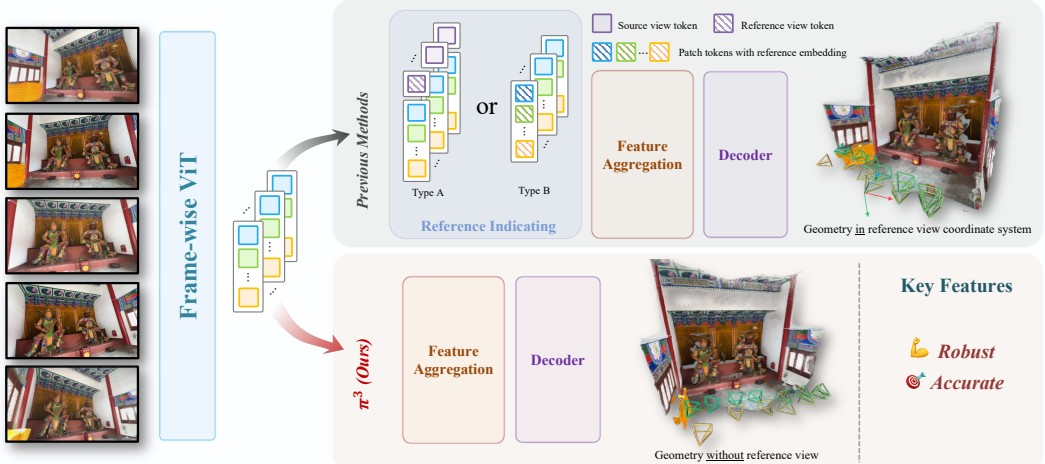

Figure 3: Unlike prior methods that designate a *reference view* by concatenating a special token (Type A) or adding a learnable embedding (Type B), $\pi^3$ achieves permutation equivariance by eliminating this requirement altogether. Instead, it employs relative supervision, making our approach inherently robust to the order of input views.

A unifying characteristic of these methods—a paradigm largely inherited from classical SfM—is their reliance on anchoring the predicted 3D structure to a designated reference frame. Our work departs from this paradigm by presenting a fundamentally different approach.

# 3  METHOD

## 3.1  PERMUTATION-EQUIVARIANT ARCHITECTURE

To ensure our model's output is invariant to the arbitrary ordering of input views, we designed our network $\phi$ to be *permutation-equivariant*.

Let the input be a sequence of $N$ images, $S = (\mathbf{I}_1, \ldots, \mathbf{I}_N)$, where each image $\mathbf{I}_i \in \mathbb{R}^{H \times W \times 3}$. The network $\phi$ maps this sequence to a corresponding tuple of output sequences:

$$\phi(S) = ((\mathbf{T}_1, \ldots, \mathbf{T}_N), (\mathbf{X}_1, \ldots, \mathbf{X}_N), (\mathbf{C}_1, \ldots, \mathbf{C}_N)) \tag{1}$$

Here, $\mathbf{T}_i \in SE(3) \subset \mathbb{R}^{4 \times 4}$ is the camera pose, $\mathbf{X}_i \in \mathbb{R}^{H \times W \times 3}$ is the associated pixel-aligned 3D point map represented in its own camera coordinate system, and $\mathbf{C}_i \in \mathbb{R}^{H \times W}$ is the confidence map of $\mathbf{X}_i$, each corresponding to the input image $\mathbf{I}_i$.

For any permutation $\pi$, let $P_\pi$ be an operator that permutes the order of a sequence. The network $\phi$ satisfies the permutation-equivariant property:

$$\phi(P_\pi(S)) = P_\pi(\phi(S)) \tag{2}$$

This means that permuting the input sequence, $P_\pi(S) = (\mathbf{I}_{\pi(1)}, \ldots, \mathbf{I}_{\pi(N)})$, results in an identically permuted output tuple:

$$P_\pi(\phi(S)) = ((\mathbf{T}_{\pi(1)}, \ldots, \mathbf{T}_{\pi(N)}), (\mathbf{X}_{\pi(1)}, \ldots, \mathbf{X}_{\pi(N)}), (\mathbf{C}_{\pi(1)}, \ldots, \mathbf{C}_{\pi(N)})) \tag{3}$$

This property guarantees a consistent one-to-one correspondence between each image and its respective output (e.g., geometry or pose). This design offers several key advantages. First, reconstruction quality becomes *independent of the reference view selection*, in contrast to prior methods that suffer from performance degradation when the reference view changes. Second, the model becomes more *robust* to uncertain or noisy observations. These claims are empirically validated in Section 4.

To realize this equivariance in practice, our implementation (illustrated in Figure 3) omits all order-dependent components, such as positional embeddings used to differentiate between frames and specialized learnable tokens that designate a reference view, like the camera tokens found in VGGT (Wang et al., 2025a). Our pipeline begins by embedding each view into a sequence of patch tokens using a DINOv2 (Oquab et al., 2023) backbone. These tokens are then processed through

a series of alternating view-wise and global self-attention layers, similar to (Wang et al., 2025a), before a final decoder generates the output. The detailed architecture of our model is provided in Appendix A.1.

## 3.2 Scale-Invariant Local Geometry

For each input image $\mathbf{I}_i$, our network predicts the geometry as a pixel-aligned 3D point map $\hat{\mathbf{X}}_i$. Each point cloud is initially defined in its own local camera coordinate system. A well-known challenge in monocular reconstruction is the inherent scale ambiguity. To address this, our network predicts the point clouds up to an unknown, yet consistent, scale factor across all $N$ images of a given scene.

Consequently, the training process requires aligning the predicted point maps, $(\hat{\mathbf{X}}_1, \ldots, \hat{\mathbf{X}}_N)$, with the corresponding ground-truth (GT) set, $(\mathbf{X}_1, \ldots, \mathbf{X}_N)$. This alignment is accomplished by solving for a single optimal scale factor, $s^*$, which minimizes the depth-weighted L1 distance across the entire image sequence. The optimization problem is formulated as:

$$s^* = \arg\min_s \sum_{i=1}^{N} \sum_{j=1}^{H \times W} \frac{1}{z_{i,j}} \|s\hat{\mathbf{x}}_{i,j} - \mathbf{x}_{i,j}\|_1 \tag{4}$$

Here, $\hat{\mathbf{x}}_{i,j} \in \mathbb{R}^3$ denotes the predicted 3D point at index $j$ of the point map $\hat{\mathbf{X}}_i$. Similarly, $\mathbf{x}_{i,j}$ is its ground-truth counterpart in $\mathbf{X}_i$. The term $z_{i,j}$ is the ground-truth depth, which is the z-component of $\mathbf{x}_{i,j}$. This problem is solved using the ROE solver proposed by (Wang et al., 2025c).

Finally, the point cloud reconstruction loss, $\mathcal{L}_{\text{points}}$, is defined using the optimal scale factor $s^*$:

$$\mathcal{L}_{\text{points}} = \frac{1}{3NHW} \sum_{i=1}^{N} \sum_{j=1}^{H \times W} \frac{1}{z_{i,j}} \|s^*\hat{\mathbf{x}}_{i,j} - \mathbf{x}_{i,j}\|_1 \tag{5}$$

To encourage the reconstruction of locally smooth surfaces, we also introduce a normal loss following Wang et al. (2025c), $\mathcal{L}_{\text{normal}}$. For each point in the predicted point map $\hat{\mathbf{X}}_i$, its normal vector $\hat{\mathbf{n}}_{i,j}$ is computed from the cross product of the vectors to its adjacent neighbors on the image grid. We then supervise these normals by minimizing the angle between them and their ground-truth counterparts $\mathbf{n}_{i,j}$:

$$\mathcal{L}_{\text{normal}} = \frac{1}{NHW} \sum_{i=1}^{N} \sum_{j=1}^{H \times W} \arccos\left(\hat{\mathbf{n}}_{i,j} \cdot \mathbf{n}_{i,j}\right) \tag{6}$$

We supervise the predicted confidence map $\mathbf{C}_i$ using a Binary Cross-Entropy (BCE) loss, denoted $\mathcal{L}_{\text{conf}}$. The ground-truth target for each point is set to 1 if its L1 reconstruction error, $\frac{1}{z_{i,j}}\|s^*\hat{\mathbf{x}}_{i,j} - \mathbf{x}_{i,j}\|_1$, is below a threshold $\epsilon$, and 0 otherwise.

## 3.3 Affine-Invariant Camera Pose

The model's permutation equivariance, combined with the inherent scale ambiguity of multi-view reconstruction, implies that the output camera poses $(\hat{\mathbf{T}}_1, \ldots, \hat{\mathbf{T}}_N)$ are only defined up to an arbitrary *similarity transformation*. This specific type of affine transformation consists of a rigid transformation and a single, unknown global scale factor.

To resolve the ambiguity of the global reference frame, we supervise the network on the relative poses between views. The predicted relative pose $\hat{\mathbf{T}}_{i \leftarrow j}$ from view $j$ to $i$ is computed as:

$$\hat{\mathbf{T}}_{i \leftarrow j} = \hat{\mathbf{T}}_i^{-1} \hat{\mathbf{T}}_j \tag{7}$$

Each predicted relative pose $\hat{\mathbf{T}}_{i \leftarrow j}$ is composed of a rotation $\hat{\mathbf{R}}_{i \leftarrow j} \in SO(3)$ and a translation $\hat{\mathbf{t}}_{i \leftarrow j} \in \mathbb{R}^3$. While the relative rotation is invariant to this global transformation, the relative translation's magnitude is ambiguous. We resolve this by leveraging the optimal scale factor, $s^*$, that is computed by aligning the predicted point map to the ground truth (as detailed in a previous section).

This single, consistent scale factor is used to rectify all predicted camera translations, allowing us to directly supervise both the rotation and the correctly-scaled translation components.

The camera loss $\mathcal{L}_{\text{cam}}$ is a weighted sum of a rotation loss term and a translation loss term, averaged over all ordered view pairs where $i \neq j$:

$$\mathcal{L}_{\text{cam}} = \frac{1}{N(N-1)} \sum_{i \neq j} \left( \mathcal{L}_{\text{rot}}(i,j) + \lambda_{trans} \mathcal{L}_{\text{trans}}(i,j) \right) \tag{8}$$

where $\lambda$ is a hyperparameter to balance the two terms.

Following Dong et al. (2025), we use angle loss for rotation and Huber loss for translation. The rotation loss minimizes the geodesic distance (angle) between the predicted relative rotation $\hat{\mathbf{R}}_{i \leftarrow j}$ and its ground-truth target $\mathbf{R}_{i \leftarrow j}$:

$$\mathcal{L}_{\text{rot}}(i,j) = \arccos \left( \frac{\text{Tr}\left( (\mathbf{R}_{i \leftarrow j})^\top \hat{\mathbf{R}}_{i \leftarrow j} \right) - 1}{2} \right) \tag{9}$$

For the translation loss, we compare our scaled prediction against the ground-truth relative translation, $\mathbf{t}_{i \leftarrow j}$. We use the Huber loss, $\mathcal{H}_\delta$, for its robustness to outliers:

$$\mathcal{L}_{\text{trans}}(i,j) = \mathcal{H}_\delta(s^* \hat{\mathbf{t}}_{i \leftarrow j} - \mathbf{t}_{i \leftarrow j}) \tag{10}$$

Furthermore, our reference-free formulation is particularly well-suited to capturing the intrinsic structure of camera trajectories. Our affine-invariant camera model builds on a key insight: real-world camera paths are highly structured, not random. They typically lie on a low-dimensional manifold—for instance, a camera orbiting an object moves along a sphere, while a car-mounted camera follows a curve.

We quantitatively analyze the structure of the predicted pose distributions in Figure 4. The eigenvalue analysis confirms that the variance of our predicted poses is concentrated along significantly fewer principal components than VGGT, validating the low-dimensional structure of our output. We discuss this further in Appendix A.3.

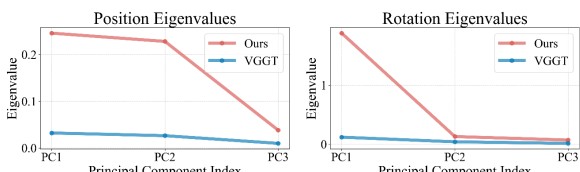

Figure 4: **Comparison of predicted pose distributions**. Our predicted pose distribution exhibits a clear low-dimensional structure.

### 3.4 MODEL TRAINING

Our model is trained end-to-end by minimizing a composite loss function, $\mathcal{L}$, which is a weighted sum of the point reconstruction loss, the confidence loss, and the camera pose loss:

$$\mathcal{L} = \mathcal{L}_{\text{points}} + \lambda_{\text{normal}} \mathcal{L}_{\text{normal}} + \lambda_{\text{conf}} \mathcal{L}_{\text{conf}} + \lambda_{\text{cam}} \mathcal{L}_{\text{cam}} \tag{11}$$

To ensure robustness and wide applicability, we train the model on a large-scale aggregation of 15 diverse datasets. This combined dataset provides extensive coverage of both indoor and outdoor environments, encompassing a wide variety of scenes from synthetic renderings to real-world captures. The specific datasets include GTA-SfM (Wang & Shen, 2020), CO3D (Reizenstein et al., 2021), WildRGB-D (Xia et al., 2024), Habitat (Savva et al., 2019), ARKitScenes (Baruch et al., 2021), TartanAir (Wang et al., 2020), ScanNet (Dai et al., 2017), ScanNet++ (Yeshwanth et al., 2023), BlendedMVG (Yao et al., 2020), MatrixCity (Li et al., 2023), MegaDepth (Li & Snavely, 2018), Hypersim (Roberts et al., 2021), Taskonomy (Zamir et al., 2018), Mid-Air (Fonder & Van Droogenbroeck, 2019), and an internal dynamic scene dataset. Details of model training can be found in Appendix A.2.

## 4 EXPERIMENTS

We report quantitative results of our method on four tasks: camera pose estimation (Sec. 4.1), point map estimation (Sec. 4.2), video depth estimation and monocular depth estimation (Sec. 4.3).

Table 1: **Camera pose estimation.** RRA, RTA, AUC are evaluated with threshold of 30 degrees.

| Method | RealEstate10K | | | Co3Dv2 (seen) | | | Sintel | | | TUM-dynamics | | | ScanNet (seen) | | |
|---|---|---|---|---|---|---|---|---|---|---|---|---|---|---|---|
| | RRA↑ | RTA↑ | AUC↑ | RRA↑ | RTA↑ | AUC↑ | ATE↓ | RPE-t↓ | RPE-r↓ | ATE↓ | RPE-t↓ | RPE-r↓ | ATE↓ | RPE-t↓ | RPE-r↓ |
| Fast3R (Yang et al., 2025) | 99.05 | 81.86 | 61.68 | 97.49 | 91.11 | 73.43 | 0.371 | 0.298 | 13.75 | 0.090 | 0.101 | 1.425 | 0.155 | 0.123 | 3.491 |
| CUT3R (Wang et al., 2025b) | 99.82 | 95.10 | 81.47 | 96.19 | 92.69 | 75.82 | 0.217 | 0.070 | 0.636 | 0.047 | 0.015 | 0.451 | 0.094 | 0.022 | 0.629 |
| FLARE (Zhang et al., 2025) | 99.69 | 95.23 | 80.01 | 96.38 | 93.76 | 73.99 | 0.207 | 0.090 | 3.015 | 0.026 | 0.013 | 0.475 | 0.064 | 0.023 | 0.971 |
| VGGT (Wang et al., 2025a) | 99.97 | 93.13 | 77.62 | 98.96 | 97.13 | 88.59 | 0.167 | 0.062 | 0.491 | 0.012 | 0.010 | 0.311 | 0.035 | 0.015 | 0.382 |
| $\pi^3$ (Ours) | 99.99 | 95.62 | 85.90 | 99.05 | 97.33 | 88.41 | 0.074 | 0.040 | 0.282 | 0.014 | 0.009 | 0.312 | 0.031 | 0.013 | 0.347 |

Across all tasks, our method achieves state-of-the-art (SOTA) or comparable performance against existing feed-forward 3D reconstruction methods. Visualized point maps are given in Figure 5 and Figure 7 (in Appendix) as qualitative results.

To validate the effectiveness of our design, We also conduct several analyses: (1) a robustness evaluation against input image sequence permutations (Sec. 4.4), (2) an ablation study on scale-invariant point maps and affine-invariant camera poses (Sec. 4.5).

## 4.1 CAMERA POSE ESTIMATION

We assess predicted camera pose using two distinct sets of metrics: angular accuracy (following (Wang et al., 2023; 2024; 2025a)) on RealEstate10K (Zhou et al., 2018) and Co3Dv2 (Reizenstein et al., 2021) datasets, and distance error (following (Zhao et al., 2022; Zhang et al., 2024; Wang et al., 2025b)) on Sintel (Bozic et al., 2021), TUM-dynamics (Sturm et al., 2012) and ScanNet (Dai et al., 2017). Details about the metrics can be found in Appendix A.5.

As shown in Table 1, our method sets a new SOTA benchmark in zero-shot generalization on Sintel and RealEstate10K, and achieves competitive SOTA results alongside VGGT on TUM-dynamics, and the in-domain Co3Dv2 and ScanNet datasets. These results underscore our model's strong generalization capabilities while maintaining excellent performance on familiar data distributions.

## 4.2 POINT MAP ESTIMATION

Following CUT3R (Wang et al., 2025b), we evaluate the quality of reconstructed multi-view point maps on the scene-level 7-Scenes (Shotton et al., 2013) and NRGBD (Azinović et al., 2022) datasets under both sparse and dense view conditions (different in sampling strides). We also extend our evaluation to the object-centric DTU (Jensen et al., 2014) and scene-level ETH3D (Schops et al., 2017) datasets. Predicted point maps are aligned to the ground truth using the Umeyama algorithm for a coarse Sim(3) alignment, followed by refinement with the Iterative Closest Point (ICP) algorithm.

Consistent with prior works (Azinović et al., 2022; Wang et al., 2024; Wang & Agapito, 2024; Wang et al., 2025b), we report Accuracy (Acc.), Completion (Comp.), and Normal Consistency (N.C.) in Table 2 and Table 3. These results highlight the strong generalization capability of our method in a broad spectrum of 3D reconstruction tasks, proving robust across synthetic and real-world scenarios, sparse and dense view settings (Table 2), as well as object-level and scene-level scales (Table 3).

Table 2: **Point map estimation on 7-Scenes and NRGBD**

| Method | View | 7-Scenes | | | | | | NRGBD | | | | | |
|---|---|---|---|---|---|---|---|---|---|---|---|---|---|
| | | Acc. ↓ | | Comp. ↓ | | NC. ↑ | | Acc. ↓ | | Comp. ↓ | | NC. ↑ | |
| | | Mean | Med. | Mean | Med. | Mean | Med. | Mean | Med. | Mean | Med. | Mean | Med. |
| Fast3R (Yang et al., 2025) | | 0.095 | 0.065 | 0.144 | 0.089 | 0.673 | 0.759 | 0.135 | 0.091 | 0.163 | 0.104 | 0.759 | 0.877 |
| CUT3R (Wang et al., 2025b) | | 0.093 | 0.049 | 0.102 | 0.051 | 0.704 | 0.805 | 0.104 | 0.041 | 0.079 | 0.031 | 0.822 | 0.968 |
| FLARE (Zhang et al., 2025) | sparse | 0.085 | 0.057 | 0.145 | 0.107 | 0.696 | 0.780 | 0.053 | 0.024 | 0.051 | 0.025 | 0.877 | 0.988 |
| VGGT (Wang et al., 2025a) | | 0.044 | 0.025 | 0.056 | 0.033 | 0.733 | 0.845 | 0.051 | 0.029 | 0.066 | 0.038 | 0.890 | 0.981 |
| $\pi^3$ (Ours) | | 0.047 | 0.029 | 0.075 | 0.049 | 0.742 | 0.841 | 0.026 | 0.015 | 0.028 | 0.014 | 0.916 | 0.992 |
| Fast3R (Yang et al., 2025) | | 0.040 | 0.017 | 0.056 | 0.018 | 0.644 | 0.725 | 0.072 | 0.030 | 0.050 | 0.016 | 0.790 | 0.934 |
| CUT3R (Wang et al., 2025b) | | 0.023 | 0.010 | 0.027 | 0.008 | 0.669 | 0.764 | 0.086 | 0.037 | 0.048 | 0.017 | 0.800 | 0.953 |
| FLARE (Zhang et al., 2025) | dense | 0.019 | 0.007 | 0.026 | 0.013 | 0.684 | 0.785 | 0.023 | 0.011 | 0.018 | 0.008 | 0.882 | 0.986 |
| VGGT (Wang et al., 2025a) | | 0.022 | 0.008 | 0.026 | 0.012 | 0.666 | 0.760 | 0.017 | 0.010 | 0.015 | 0.005 | 0.893 | 0.988 |
| $\pi^3$ (Ours) | | 0.016 | 0.007 | 0.022 | 0.011 | 0.689 | 0.792 | 0.015 | 0.008 | 0.013 | 0.005 | 0.898 | 0.987 |

Table 3: **Point map estimation on DTU and ETH3D**

| Method | DTU | | | | | | ETH3D | | | | | |
|---|---|---|---|---|---|---|---|---|---|---|---|---|
| | Acc. ↓ | | Comp. ↓ | | N.C. ↑ | | Acc. ↓ | | Comp. ↓ | | N.C. ↑ | |
| | Mean | Med. | Mean | Med. | Mean | Med. | Mean | Med. | Mean | Med. | Mean | Med. |
| Fast3R (Yang et al., 2025) | 3.340 | 1.919 | 2.929 | 1.125 | 0.671 | 0.755 | 0.832 | 0.691 | 0.978 | 0.683 | 0.667 | 0.766 |
| CUT3R (Wang et al., 2025b) | 4.742 | 2.600 | 3.400 | 1.316 | 0.679 | 0.764 | 0.617 | 0.525 | 0.747 | 0.579 | 0.754 | 0.848 |
| FLARE (Zhang et al., 2025) | 2.541 | 1.468 | 3.174 | 1.420 | 0.684 | 0.774 | 0.464 | 0.338 | 0.664 | 0.395 | 0.744 | 0.864 |
| VGGT (Wang et al., 2025a) | 1.338 | 0.779 | 1.896 | 0.992 | 0.676 | 0.766 | 0.280 | 0.185 | 0.305 | 0.182 | 0.853 | 0.950 |
| $\pi^3$ (Ours) | 1.198 | 0.646 | 1.849 | 0.607 | 0.678 | 0.768 | 0.194 | 0.131 | 0.210 | 0.128 | 0.883 | 0.969 |

Table 4: **Video depth estimation on Sintel, Bonn and KITTI.** FPS is evaluated on KITTI using one A800 GPU.

| Method | Params | Sintel | | Bonn | | KITTI | | FPS |
|---|---|---|---|---|---|---|---|---|
| | | Abs Rel ↓ | $\delta < 1.25$ ↑ | Abs Rel ↓ | $\delta < 1.25$ ↑ | Abs Rel ↓ | $\delta < 1.25$ ↑ | |
| DUSt3R (Wang et al., 2024) | 571M | 0.662 | 0.434 | 0.151 | 0.839 | 0.143 | 0.814 | 1.25 |
| MASt3R (Leroy et al., 2024) | 689M | 0.558 | 0.487 | 0.188 | 0.765 | 0.115 | 0.848 | 1.01 |
| MonST3R (Zhang et al., 2024) | 571M | 0.399 | 0.519 | 0.072 | 0.957 | 0.107 | 0.884 | 1.27 |
| Fast3R (Yang et al., 2025) | 648M | 0.638 | 0.422 | 0.194 | 0.772 | 0.138 | 0.834 | **65.8** |
| MVDUSt3R (Tang et al., 2024) | 661M | 0.805 | 0.283 | 0.426 | 0.357 | 0.456 | 0.342 | 0.69 |
| CUT3R (Wang et al., 2025b) | 793M | 0.417 | 0.507 | 0.078 | 0.937 | 0.122 | 0.876 | 6.98 |
| Aether (Team et al., 2025) | 5.57B | 0.324 | 0.502 | 0.273 | 0.594 | 0.056 | 0.978 | 6.14 |
| FLARE (Zhang et al., 2025) | 1.40B | 0.729 | 0.336 | 0.152 | 0.790 | 0.356 | 0.570 | 1.75 |
| VGGT (Wang et al., 2025a) | 1.26B | 0.299 | 0.638 | 0.057 | 0.966 | 0.062 | 0.969 | 43.2 |
| $\pi^3$ (Ours) | 959M | **0.233** | **0.664** | **0.049** | **0.975** | **0.038** | **0.986** | 57.4 |

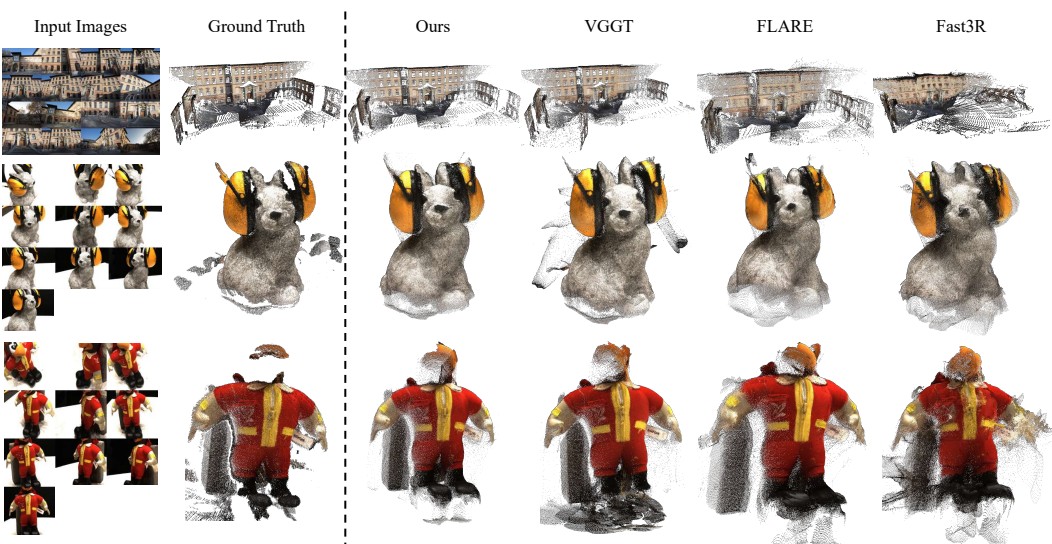

Figure 5: **Qualitative comparison of multi-view 3D reconstruction.** Compared to other multi-frame feed-forward reconstruction methods, $\pi^3$ produces cleaner, more accurate and more complete reconstructions with fewer artifacts.

## 4.3 DEPTH ESTIMATION

Following the methodology of CUT3R (Wang et al., 2025b), we report the Absolute Relative Error (Abs Rel) and the prediction accuracy at a threshold of $\delta < 1.25$ of our method on the tasks of video depth estimation and monocular depth estimation, using the Sintel (Bozic et al., 2021), Bonn (Palazzolo et al., 2019), and KITTI (Geiger et al., 2013) datasets. NYU-v2 Silberman et al. (2012) is additionally used for monocular depth estimation.

**Video depth estimation.** In this setting, video depth sequences are aligned to the ground truth with a scale per sequence. As reported in Table 4, our method achieves a new SOTA performance across all three datasets within feed-forward 3D reconstruction methods. Notably, it also delivers exceptional efficiency, running at 57.4 FPS on KITTI, significantly faster than VGGT (43.2 FPS) and Aether (6.14 FPS), despite having a smaller model size.

**Monocular depth estimation.** In this setting, each depth map is aligned independently to its ground truth with a scale factor. As reported in Table 5, our method achieves state-of-the-art results among multi-frame feed-forward reconstruction approaches, even though it is not explicitly optimized for single-frame depth estimation. Meanwhile, it performs competitively with MoGe (Wang et al., 2025c;d), one of the top-performing monocular depth estimation models.

## 4.4 ROBUSTNESS EVALUATION

A key property of our proposed architecture is permutation equivariance, ensuring that its outputs are robust to variations in the input image sequence order. To empirically verify this, we conduct

Table 5: **Monocular depth estimation**

| Method | Sintel | | Bonn | | KITTI | | NYU-v2 | |
|---|---|---|---|---|---|---|---|---|
| | Abs Rel↓ | $\delta < 1.25$ ↑ | Abs Rel↓ | $\delta < 1.25$ ↑ | Abs Rel↓ | $\delta < 1.25$ ↑ | Abs Rel↓ | $\delta < 1.25$ ↑ |
| DUSt3R (Wang et al., 2024) | 0.488 | 0.532 | 0.139 | 0.832 | 0.109 | 0.873 | 0.081 | 0.909 |
| MASt3R (Leroy et al., 2024) | 0.413 | 0.569 | 0.123 | 0.833 | 0.077 | 0.948 | 0.110 | 0.865 |
| MonST3R (Zhang et al., 2024) | 0.402 | 0.525 | 0.069 | 0.954 | 0.098 | 0.895 | 0.094 | 0.887 |
| Fast3R (Yang et al., 2025) | 0.544 | 0.509 | 0.169 | 0.796 | 0.120 | 0.861 | 0.093 | 0.898 |
| CUT3R (Wang et al., 2025b) | 0.418 | 0.520 | 0.058 | 0.967 | 0.097 | 0.914 | 0.081 | 0.914 |
| FLARE (Zhang et al., 2025) | 0.606 | 0.402 | 0.130 | 0.836 | 0.312 | 0.513 | 0.089 | 0.898 |
| VGGT (Wang et al., 2025a) | 0.335 | 0.599 | 0.053 | 0.970 | 0.082 | 0.947 | 0.056 | 0.951 |
| MoGe | **0.273** | **0.695** | 0.050 | 0.976 | **0.049** | **0.979** | 0.055 | 0.952 |
| − v1 (Wang et al., 2025c) | − **0.273** | − **0.695** | − 0.050 | − 0.976 | − 0.054 | − 0.977 | − 0.055 | − 0.952 |
| − v2 (Wang et al., 2025d) | − 0.277 | − 0.687 | − 0.063 | − 0.973 | − **0.049** | − **0.979** | − 0.060 | − 0.940 |
| $\pi^3$ (Ours) | 0.277 | 0.614 | **0.044** | **0.976** | 0.060 | 0.971 | **0.054** | **0.956** |

Table 6: **Standard deviation of point cloud estimation**

| Method | DTU | | | | | | ETH3D | | | | | |
|---|---|---|---|---|---|---|---|---|---|---|---|---|
| | Acc. std. ↓ | | Comp. std. ↓ | | N.C. std. ↓ | | Acc. std. ↓ | | Comp. std. ↓ | | N.C. std. ↓ | |
| | Mean | Med. | Mean | Med. | Mean | Med. | Mean | Med. | Mean | Med. | Mean | Med. |
| Fast3R (Yang et al., 2025) | 0.578 | 0.451 | 0.677 | 0.376 | 0.007 | 0.009 | 0.182 | 0.205 | 0.381 | 0.273 | 0.047 | 0.072 |
| FLARE (Zhang et al., 2025) | 0.720 | 0.494 | 1.346 | 1.134 | 0.009 | 0.012 | 0.171 | 0.187 | 0.251 | 0.188 | 0.048 | 0.053 |
| VGGT (Wang et al., 2025a) | 0.033 | 0.022 | 0.054 | 0.036 | 0.007 | 0.007 | 0.049 | 0.040 | 0.062 | 0.042 | 0.022 | 0.015 |
| $\pi^3$ (Ours) | **0.003** | **0.002** | **0.006** | **0.003** | **0.001** | **0.001** | **0.000** | **0.000** | **0.000** | **0.000** | **0.001** | **0.000** |

experiments on the DTU (Jensen et al., 2014) and ETH3D (Schops et al., 2017) datasets. For each sequence of length N, we create N different input orderings, by making each of the N frames the first frame in the sequence in turn. We then compute the standard deviation of the metrics across these N runs. We then compute the standard deviation of the reconstruction metrics across these $N$ outputs. A lower standard deviation indicates higher robustness to input order variations.

As reported in Table 4.4, our method achieves near-zero standard deviation across all metrics on DTU and ETH3D, outperforming existing approaches by several orders of magnitude. For instance, on DTU, our mean accuracy standard deviation is 0.003, while VGGT reports 0.033. On ETH3D, our model achieves effectively zero variance. This stark contrast highlights the limitations of reference-frame-dependent methods, which exhibit significant sensitivity to input order. Our results provide compelling evidence that the proposed architecture is genuinely permutation-equivariant, ensuring consistent and order-independent 3D reconstruction.

## 4.5 ABLATION STUDY

To validate the effectiveness of our proposed components, we conducted an ablation study by systematically removing features from our complete model. We define two ablated variants of our full model: Model 2, which lacks the affine-invariant camera pose modeling, and Model 1, which lacks both affine-invariant poses and scale-invariant pointmaps. See Appendix A.6 for more details.

The comparative results for pointmap estimation across three datasets are presented in Table 7. We found that scale-invariant pointmap modeling does not yield significant performance gains on indoor datasets like 7-Scenes and NRGBD. For outdoor data, however, the performance improvement is substantially more pronounced. This observation is consistent with previous studies on scale-invariant depth, which have shown that outdoor scenes are more significantly affected by scale ambiguity. Furthermore, we observed that affine-invariant camera pose modeling consistently enhances the final performance. More importantly, unlike Model 1 and Model 2, its inclusion renders the model permutation-equivariant. Consequently, the model becomes robust to both the order of input frames and the selection of the reference view.

Table 7: **Ablation study on the key components of our model.** We show how the performance metric improves as each component is added to the baseline.

| Model | ETH3D | | | | | | 7-Scenes | | | | | | NRGBD | | | | | |
|---|---|---|---|---|---|---|---|---|---|---|---|---|---|---|---|---|---|---|
| | Acc. ↓ | | Comp. ↓ | | N.C ↑ | | Acc. ↓ | | Comp. ↓ | | N.C ↑ | | Acc. ↓ | | Comp. ↓ | | N.C ↑ | |
| | Mean | Med. | Mean | Med. | Mean | Med. | Mean | Med. | Mean | Med. | Mean | Med. | Mean | Med. | Mean | Med. | Mean | Med. |
| Model 1 | 0.229 | 0.150 | 0.166 | 0.103 | 0.802 | 0.930 | 0.020 | 0.010 | **0.019** | 0.009 | 0.715 | 0.834 | 0.034 | 0.018 | 0.025 | 0.011 | 0.859 | 0.977 |
| Model 2 | 0.197 | 0.118 | 0.118 | 0.065 | 0.820 | 0.943 | 0.020 | **0.009** | 0.020 | **0.008** | 0.716 | 0.837 | 0.031 | 0.018 | 0.023 | **0.010** | 0.861 | 0.978 |
| Full Model | **0.131** | **0.076** | **0.079** | **0.043** | **0.841** | **0.957** | **0.019** | **0.009** | 0.020 | 0.009 | **0.723** | **0.843** | **0.028** | **0.015** | **0.022** | **0.010** | **0.875** | **0.981** |

## 5 CONCLUSION

In this work, we introduced $\pi^3$, a feed-forward neural network that presents a new paradigm for visual geometry reconstruction by eliminating the reliance on a fixed reference view. By leveraging a fully permutation-equivariant architecture, our model is inherently robust to input ordering and leads to higher accuracy. This design choice removes a critical inductive bias found in previous methods, allowing our simple yet powerful approach to achieve state-of-the-art performance on a wide array of tasks, including camera pose estimation, depth estimation, and dense reconstruction. $\pi^3$ demonstrates that reference-free systems are not only viable but can lead to more stable and versatile 3D vision models.

ACKNOWLEDGMENTS

This work is supported by Shanghai Artificial Intelligence Laboratory.

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

## A  APPENDIX

### A.1  ARCHITECTURE DETAILS

The encoder and alternating attention modules are the same as those in VGGT (Wang et al., 2025a), with the exception that we use only 36 layers for the alternating attention module, whereas VGGT uses 48. The decoders for camera poses, local point maps, and confidence scores share the same architecture but do not share weights. This architecture is a lightweight, 5-layer transformer that applies self-attention exclusively to the features of each individual image. Following the decoder, the output heads vary by task. The heads for local point maps and confidence scores consist of a simple MLP followed by a pixel shuffle operation. For camera poses, the head is adapted from Reloc3r (Dong et al., 2025) and uses an MLP, average pooling, and another MLP. The rotation is initially predicted in a 9D representation (Levinson et al., 2020) and is then converted to a 3×3 rotation matrix via SVD orthogonalization.

### A.2  TRAINING DETAILS

We train $\pi^3$ in two stages, a process similar to Dust3R (Wang et al., 2024). First, the model is trained on a low resolution of 224 × 224 pixels. Then, it is fine-tuned on images of random resolutions where the total pixel count is between 100,000 and 255,000 and the aspect ratio is sampled from the range [0.5, 2.0], a strategy similar to MoGe (Wang et al., 2025c). We use a dynamic batch sizing strategy similar to VGGT. In the first stage, we sample 64 images per GPU, and in the second stage, we sample 48 images per GPU. Each batch is composed of

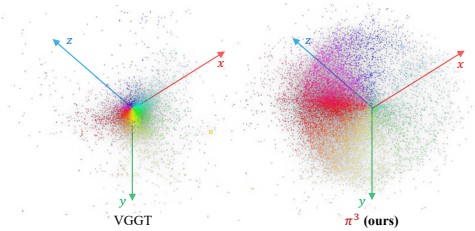

Figure 6: **Comparison of predicted pose distributions**. We visualize the predicted pose distributions in 3D space. $\pi^3$ shows a clear low-dimensional structure, while VGGT's distribution is scattered.

Table 8: **Comparison with VGGT when trained from scratch.**

| Method | ETH3D | | 7-Scenes | | NRGB | |
|---|---|---|---|---|---|---|
| | Acc. ↓ | Comp. ↓ | Acc. ↓ | Comp. ↓ | Acc. ↓ | Comp. ↓ |
| $\pi^3$ | 0.618 | 0.453 | 0.064 | 0.068 | 0.071 | 0.047 |
| VGGT (Wang et al., 2025a) | 0.563 | 0.449 | **0.057** | **0.046** | 0.060 | 0.042 |
| $\pi^3$ + global proxy | **0.418** | **0.266** | 0.059 | 0.071 | **0.052** | **0.035** |

2 to 24 images. Each training stage runs for 80
epochs, with each epoch comprising 800 iterations. Our final model is not trained from scratch. Instead, we initialize the weights for the encoder and the alternating attention module from the pre-trained VGGT model, and we keep the encoder frozen during training. We train the first stage on 16 A100 GPUs and the second stage on 64 A100 GPUs. For our loss function, we set the weights for each component as follows: $\lambda_{normal} = 1.0$, $\lambda_{conf} = 0.05$, $\lambda_{cam} = 0.1$, and $\lambda_{trans} = 100.0$. The implementation of our normal loss follows that of MoGe, and the resolution for aligning the local point map loss is set to 4096. Regarding optimization, we set the initial learning rate for all model components to $5 \times 10^{-5}$. We employ a `OneCycleLR` scheduler, where the learning rate anneals from its maximum value down to a minimal value over the entire training duration following a cosine curve. We use the same learning rate and scheduler settings for both stages. The confidence head is not trained jointly with the other modules. Instead, after completing the two main training stages, we freeze the rest of the network and train the confidence head in isolation. This final stage converges rapidly, typically within a few epochs, without impacting the model's overall performance. We use gradient clipping with a norm of 1.0.

### A.3   DISCUSSION FOR PREDICTED POSE DISTRIBUTION

In Figure 6, we analyze the geometric properties of the learned representations by visualizing the distribution of predicted camera poses. In this plot, the spatial coordinates $(x, y, z)$ correspond to the translation component, while the rotation is encoded into the RGB color space. Specifically, we convert each predicted rotation matrix into an axis-angle vector, normalize its components to the range $[0, 1]$, and map them to the Red, Green, and Blue channels. The visualization reveals a striking contrast: while VGGT's distribution appears scattered and random, our predictions form a distinct low-dimensional structure. This suggests that our model effectively captures the underlying geometric manifold, which is likely a key factor contributing to its superior performance.

### A.4   COMPARISON WITH VGGT

This section details an experiment designed solely for a fair comparison against VGGT (Wang et al., 2025a). A direct comparison is challenging because training our model *from scratch* with only its core objectives (camera poses and local pointmaps) leads to suboptimal convergence, whereas VGGT's design incorporates a multi-task learning setup.

We attribute this difficulty to the "cold start" problem inherent in relative pose supervision. Unlike reference-anchored methods, our approach generates highly coupled $N \times N$ relative constraints, which are significantly more unstable to optimize from a completely random initialization.

To address this, we introduce an auxiliary head to predict a global pointmap relative to a reference frame, using a loss analogous to Eq. 3.2. Crucially, while the reference view is used via cross-attention in this head, it serves purely as a *proxy task* to decouple geometry learning and stabilize the optimization landscape. Our final model remains fully permutation-equivariant.

We train both our adapted model and VGGT under these identical, multi-task conditions: *from scratch* (except for DINOv2 encoders) on the same data, at a $224 \times 224$ resolution for 80 epochs (800 steps/epoch). We use the same data as described in Section 3.4.

As shown in Table 8, once the optimization stability is ensured by the global proxy, $\pi^3$ significantly outperforms the VGGT baseline on ETH3D and NRGB benchmarks. Note that while our model can be trained from scratch effectively with this proxy, we utilize VGGT initialization in our main experiments to maximize computational efficiency and leverage the large-scale data priors captured in the pre-trained weights.

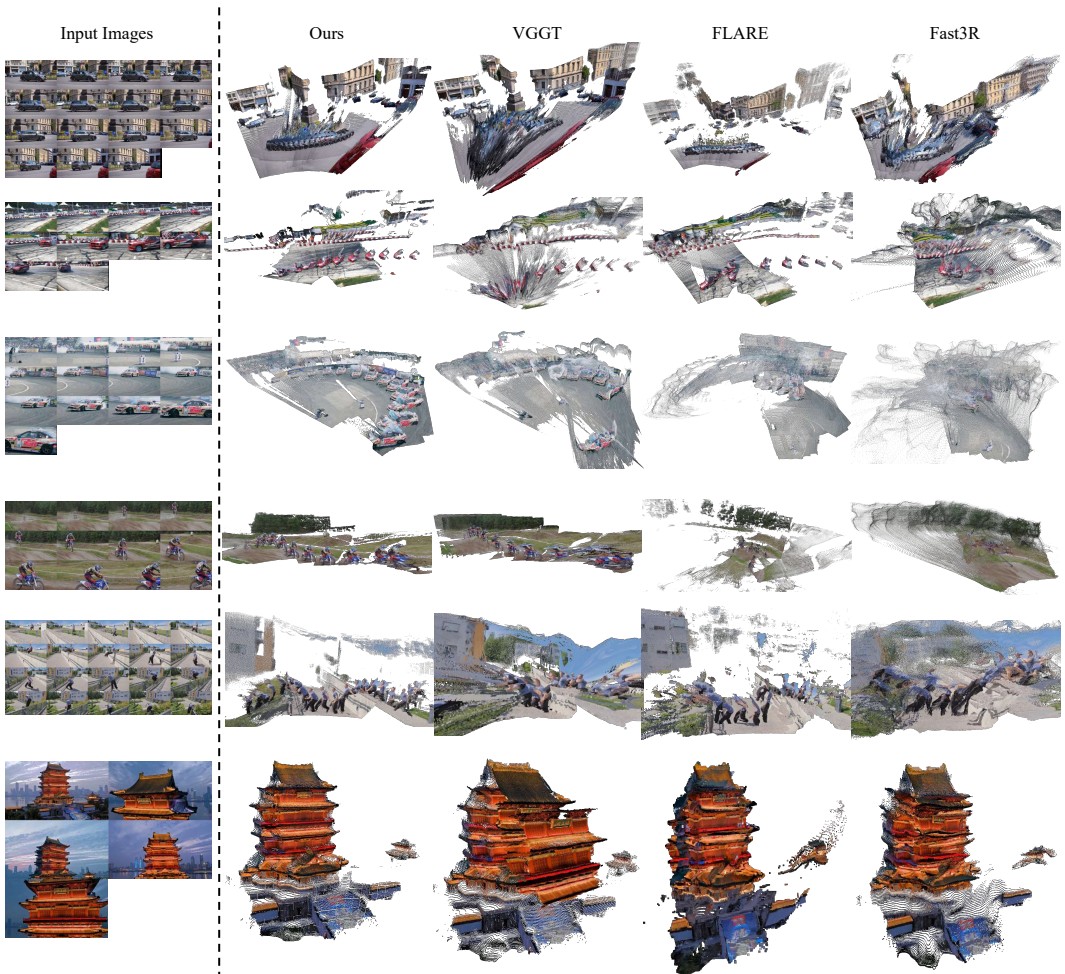

Figure 7: **Qualitative comparison of in-the-wild multi-view 3D reconstruction.** $\pi^3$ demonstrates superior robustness on challenging in-the-wild sequences, consistently producing more coherent and complete 3D structures for both dynamic and complex static scenes compared to other feed-forward approaches.

## A.5  CAMERA POSE EVALUATION METRICS

**Angular Accuracy Metrics.**   Following prior work (Wang et al., 2024; 2025a), we evaluate predicted camera poses on the scene-level RealEstate10K (Zhou et al., 2018) and object-centric Co3Dv2 (Reizenstein et al., 2021) datasets, both featuring over 1000 test sequences. For each sequence, we randomly sample 10 images, form all possible pairs, and compute the angular errors of the relative rotation and translation vectors. This process yields the Relative Rotation Accuracy (RRA) and Relative Translation Accuracy (RTA) at a given angular threshold (e.g. 30 degrees). The Area Under the Curve (AUC) of the min(RRA,RTA)-threshold curve serves as a unified metric. All methods in Table 1 have been trained on Co3Dv2, while RealEstate10K is excluded from trainset except for CUT3R (Wang et al., 2025b).

**Distance Error Metrics.** Following (Wang et al., 2025b), we report the Absolute Trajectory Error (ATE), Relative Pose Error for translation (RPE-t), and Relative Pose Error for rotation (RPE-r) on the synthetic outdoor Sintel (Bozic et al., 2021) dataset, as well as the real-world indoor TUM-dynamics (Sturm et al., 2012) and ScanNet (Dai et al., 2017) datasets. Predicted camera trajectories are aligned with the ground truth via a Sim(3) transformation before calculating the errors. All methods in Table 1 have seen ScanNet or ScanNet++ (Yeshwanth et al., 2023) samples during training time. Zero-shot pose estimation accuracy is evaluated on Sintel and TUM-dynamics for all methods.

## A.6 ABLATION DETAILS

The primary difference between our full model and the ablated models (Model 1 and Model 2) is that the latter two incorporate a camera token. This token is essential for distinguishing the reference view, as the model is no longer permutation-equivariant after the removal of the affine-invariant camera pose modeling. At each iteration, the camera token is concatenated with a randomly selected reference view before the alternating-attention module similar to (Wang et al., 2025a). We compute an angle loss for rotation and a Huber loss for translation between the predicted and ground-truth poses in the reference view's coordinate system for Model 1 and Model 2. While Model 1 and Model 2 share an identical architecture and parameter count, their key distinctions lie in the loss calculation and normalization processes. For Model 1, we neither perform alignment during the loss computation for the predicted pointmap nor do we normalize the pointmap itself. We found that applying normalization in this specific case led to anomalous and significantly degraded performance, a phenomenon also observed in prior work (Wang et al., 2025a). In contrast, the predicted local pointmaps are normalized for both Model 2 and the full model.

For a fair comparison, all models were trained for 80 epochs, with 800 iterations per epoch, on images with a resolution of $224 \times 224$. They shared the same initialization procedure as our final model: we loaded pre-trained weights for the VGGT encoder and alternating-attention layers, and kept the encoder frozen throughout training. For the 7-Scenes and NRGBD datasets, we use the same dense view setting as in the previous section.

## A.7 ADDITIONAL EVALUATION

**Camera pose estimation** with tighter angular thresholds. Following the protocol of VGGT (Wang et al., 2025a), Tab. 1 primarily reports the RRA, RTA, and AUC metrics using a relaxed angular threshold of $30°$. However, to better assess precision, we also examine tighter thresholds, such as $5°$ and $15°$ used by Fast3R (Yang et al., 2025) and FLARE (Zhang et al., 2025). Accordingly, in Tab. 9, we present a full set of RRA, RTA, and AUC metrics across thresholds of $1°$, $3°$, $5°$, $10°$, and $15°$, evaluated on RealEstate10K. Our $\pi^3$ model demonstrates robust and consistent performance even with these more demanding constraints.

Table 9: **Camera pose estimation with tighter angular thresholds on RealEstate10K**

| Method | RRA (↑) | | | | | RTA (↑) | | | | | AUC (↑) | | | | |
|---|---|---|---|---|---|---|---|---|---|---|---|---|---|---|---|
| | @1 | @3 | @5 | @10 | @15 | @1 | @3 | @5 | @10 | @15 | @1 | @3 | @5 | @10 | @15 |
| Fast3R (Yang et al., 2025) | 54.30 | 87.24 | 94.78 | 97.90 | 98.46 | 5.47 | 24.56 | 39.23 | 59.29 | 69.11 | 3.77 | 13.67 | 22.36 | 37.33 | 46.71 |
| CUT3R (Wang et al., 2025b) | 78.63 | 96.06 | 98.15 | 99.31 | 99.63 | 16.23 | 51.43 | 67.44 | 82.98 | 88.93 | 13.40 | 33.39 | 45.63 | 61.78 | 70.15 |
| FLARE (Zhang et al., 2025) | 70.99 | 93.42 | 97.11 | 98.98 | 99.44 | 11.01 | 43.33 | 62.39 | 82.29 | 89.20 | 8.43 | 25.67 | 38.47 | 57.20 | 67.02 |
| VGGT (Wang et al., 2025a) | 69.68 | 92.70 | 97.06 | 99.40 | 99.74 | 8.58 | 39.93 | 60.61 | 80.20 | 86.34 | 6.23 | 22.25 | 35.46 | 54.76 | 64.54 |
| $\pi^3$ (Ours) | **85.19** | **97.56** | **98.83** | **99.63** | **99.86** | **27.57** | **65.57** | **78.32** | **88.69** | **92.02** | **24.87** | **47.28** | **58.63** | **72.11** | **78.39** |

**Point map estimation** with Chamfer Distance(CD). To further evaluate the quality of the point map estimation, we additionaly calculate the Chamfer Distance metric, which is defined as the mean value of the Accuracy (Acc.) and Completion (Comp.) terms. The results across all evaluation datasets are reported in Tab. 10.

Table 10: **Point map estimation with Chamfer Distance.**

| Method | 7-Scenes-sparse | | 7-Scenes-dense | | NRGBD-sparse | | NRGBD-dense | | DTU | | ETH3D | |
|---|---|---|---|---|---|---|---|---|---|---|---|---|
| | CD-mean↓ | CD-med↓ | CD-mean↓ | CD-med↓ | CD-mean↓ | CD-med↓ | CD-mean↓ | CD-med↓ | CD-mean↓ | CD-med↓ | CD-mean↓ | CD-med↓ |
| Fast3R (Yang et al., 2025) | 0.150 | 0.111 | 0.048 | 0.018 | 0.150 | 0.097 | 0.061 | 0.024 | 3.134 | 1.476 | 0.875 | 0.646 |
| CUT3R (Wang et al., 2025b) | 0.097 | 0.049 | 0.025 | **0.009** | 0.091 | 0.036 | 0.065 | 0.025 | 4.021 | 1.886 | 0.684 | 0.551 |
| FLARE (Zhang et al., 2025) | 0.115 | 0.083 | **0.023** | 0.010 | 0.052 | 0.023 | 0.020 | 0.009 | 2.834 | 1.409 | 0.564 | 0.377 |
| VGGT (Wang et al., 2025a) | **0.050** | **0.029** | 0.024 | 0.010 | 0.058 | 0.032 | 0.015 | 0.007 | 1.619 | 0.888 | 0.287 | 0.177 |
| $\pi^3$ (Ours) | 0.061 | 0.039 | **0.019** | **0.009** | **0.026** | **0.013** | **0.013** | **0.006** | **1.472** | **0.626** | **0.199** | **0.128** |

**Monocular depth estimation** compared with Depth Anything V2 (Yang et al., 2024), one of the SOTA models for monocular depth estimation. We evaluate it on our standard benchmarks with input resolution 518, following CUT3R (Wang et al., 2025b) protocol. As shown in Tab. 11, $\pi^3$ achieves comparable performance to the specialized DAv2, despite being designed for generalist multi-view reconstruction.

Table 11: **Monocular depth estimation.**

| Method | Sintel | | Bonn | | KITTI | | NYU-v2 | |
|---|---|---|---|---|---|---|---|---|
| | Abs Rel↓ | $\delta < 1.25$ ↑ | Abs Rel↓ | $\delta < 1.25$ ↑ | Abs Rel↓ | $\delta < 1.25$ ↑ | Abs Rel↓ | $\delta < 1.25$ ↑ |
| DA V2 (Yang et al., 2024) | 0.372 | 0.541 | 0.126 | 0.804 | 0.090 | 0.919 | 0.081 | 0.921 |
| – metric indoor | – 0.372 | – 0.541 | – 0.126 | – 0.804 | – 0.097 | – 0.912 | – 0.081 | – 0.921 |
| – metric outdoor | – 0.478 | – 0.477 | – 0.186 | – 0.668 | – 0.090 | – 0.919 | – 0.172 | – 0.689 |
| MoGe | **0.273** | **0.695** | 0.050 | 0.976 | **0.049** | **0.979** | 0.055 | 0.952 |
| – v1 (Wang et al., 2025c) | – **0.273** | – **0.695** | – 0.050 | – 0.976 | – 0.054 | – 0.977 | – 0.055 | – 0.952 |
| – v2 (Wang et al., 2025d) | – 0.277 | – 0.687 | – 0.063 | – 0.973 | – **0.049** | – **0.979** | – 0.060 | – 0.940 |
| $\pi^3$ (Ours) | 0.277 | 0.614 | **0.044** | **0.976** | 0.060 | 0.971 | **0.054** | **0.956** |

## A.8 LIMITATIONS

Our model demonstrates strong performance, but it also has several key limitations. First, it is unable to handle transparent objects, as our model does not explicitly account for complex light transport phenomena. Second, compared to contemporary diffusion-based approaches, our reconstructed geometry lacks the same level of fine-grained detail. Finally, the point cloud generation relies on a simple upsampling mechanism using an MLP with pixel shuffling. While efficient, this design can introduce noticeable grid-like artifacts, particularly in regions with high reconstruction uncertainty.

## A.9 LLM USAGE STATEMENT

In the preparation of this manuscript, we utilized a Large Language Model (LLM) as a writing assistant. The LLM's role was strictly limited to improving the manuscript's clarity, correcting grammatical errors, and refining the overall language for professional academic standards. All scientific contributions, including the core ideas, methodology, experimental design, and interpretation of results, are the original work of the authors.

