# OpenReview forum: "$\pi^3$: Permutation-Equivariant Visual Geometry Learning"
_ICLR.cc/2026/Conference — ICLR 2026 Poster_

### Official Review · Reviewer_DanB · 2025-10-24

**Soundness:** 4
**Presentation:** 4
**Contribution:** 3
**Rating:** 8
**Confidence:** 4

**Summary:**

This paper introduces a novel permutation equivariant architecture for feed-forward 3D reconstruction. To achieve permutation equivariance, positional embeddings used to differentiate between frames and learnable tokens that denote a reference view are not used. The structure is straightforward, while the performance is impressive on various tasks.

**Strengths:**

1. The paper is well-written and easy to follow.

2. The paper addresses the limitation of existing feed-forward 3D reconstruction pipelines, where one view is selected as a reference view and the performance may degrade if the reference view changes.

3. The performance in camera pose estimation, monocular/video depth estimation, and pointmap reconstruction, is state-of-the-art.

4. Ablation study verifies the permutation equivariance of Pi3.

**Weaknesses:**

1. Camera pose estimation: In addition to using large angular threshold (30 degrees), more strict thresholds, e.g. 5 degrees, 10 degrees, should be used to illustrate the rotation accuracy.

2. Point-map evaluation:
    * On both DTU and ETH3D, the numbers of VGGT are very different from those in the original paper. Could the authors explain the reason?
    * Since the authors already provide accuracy and completeness, it makes sense to further add Chamfer distance/F-score, which is the best metric to evaluate the overall reconstruction quality.

3. In Table 6, CUT3R heavily relies on the sequential/temporal information. Thus, it is not a very good baseline here.

**Questions:**

1. To compute the scale $s$ that aligns ground truth and prediction, the other solution is to use umeyama to align the ground truth and predicted camera poses. Different form optimizing Eq. 4, this solution looks simpler. Did the authors try this?

---

> ### Author Response · Authors · 2025-11-21
>
> We are grateful for the "Accept" recommendation and the encouraging feedback. We appreciate that you recognized the novelty of our permutation-equivariant architecture and its effectiveness in addressing the reference view limitation. Below, we address your specific questions and suggestions.
>
> ### **1. Evaluation on Stricter Camera Pose Thresholds**
> > *In addition to using large angular threshold (30 degrees), more strict thresholds, e.g. 5 degrees, 10 degrees, should be used...*
>
> We agree that stricter thresholds provide a more comprehensive view of precision. We have computed the AUC metrics at $5^\circ$ and $10^\circ$ thresholds on RealEstate10K. As shown below, $\pi^3$ maintains robust performance even under stricter regimes:
>
> | Method | Racc_1↑ | Tacc_1↑ | Auc_1↑ | Racc_3↑ | Tacc_3↑ | Auc_3↑ | Racc_5↑ | Tacc_5↑ | Auc_5↑ | Racc_10↑ | Tacc_10↑ | Auc_10↑ |
> | :--- | :---: | :---: | :---: | :---: | :---: | :---: | :---: | :---: | :---: | :---: | :---: | :---: |
> | fast3r | 54.30 | 5.47 | 3.77 | 87.24 | 24.56 | 13.67 | 94.78 | 39.23 | 22.36 | 97.90 | 59.29 | 37.33 |
> | flare | 70.99 | 11.01 | 8.43 | 93.42 | 43.33 | 25.67 | 97.11 | 62.39 | 38.47 | 98.98 | 82.29 | 57.20 |
> | cut3r | 78.63 | 16.23 | 13.40 | 96.06 | 51.43 | 33.39 | 98.15 | 67.44 | 45.63 | 99.31 | 82.98 | 61.78 |
> | vggt | 69.68 | 8.58 | 6.23 | 92.70 | 39.93 | 22.25 | 97.06 | 60.61 | 35.46 | 99.40 | 80.20 | 54.76 |
> | pi3 | 85.19 | 27.57 | 24.87 | 97.56 | 65.57 | 47.28 | 98.83 | 78.32 | 58.63 | 99.63 | 88.69 | 72.11 |
>
> ### 2. **Point-map Evaluation and Metrics**
> >  *Discrepancy in VGGT numbers:*
>
> The discrepancy arises because VGGT has not released official evaluation code. To ensure a strictly fair comparison, we re-evaluated VGGT using a standardized protocol for all methods. Specifically:
> - Data Processing: We followed the preprocessing protocols of Spann3R for DTU, and used official ETH3D with RGB and depth undistorted.
> - Evaluation Code: We adopted the evaluation framework from CUT3R (Sim3 alignment + ICP refinement). We believe these unified settings provide a more accurate reflection of relative model performance than numbers taken directly from papers with differing protocols.
>
> > *Adding Chamfer Distance/F-score*
>
> We agree that Chamfer Distance (CD) and F-score are excellent metrics for overall quality. We have added CD for 7scenes, NRGBD, DTU and ETH3D datasets below (Input resolution: 518):
>
> | model  | 7scenes CD-mean↓ | 7scenes CD-med↓ | NRGBD CD-mean↓ | NRGBD CD-med↓ | DTU CD-mean↓ | DTU CD-med↓ | ETH3D CD-mean↓ | ETH3D CD-med↓ |
> | :----- | :-----------------: | :----------------: | :---------------: | :--------------: | :-------------: | :------------: | :---------------: | :--------------: |
> | fast3r |        0.150        |       0.111        |       0.150       |      0.097       |      3.134      |     1.476      |       0.875       |      0.646       |
> | cut3r  |        0.097        |       0.049        |       0.091       |      0.036       |      4.021      |     1.886      |       0.684       |      0.551       |
> | flare  |        0.115        |       0.083        |       0.052       |      0.023       |      2.834      |     1.409      |       0.564       |      0.377       |
> | vggt   |        0.050        |       0.029        |       0.058       |      0.032       |      1.619      |     0.888      |       0.287       |      0.177       |
> | pi3    |        0.061        |       0.039        |       0.026       |      0.013       |      1.472      |     0.626      |       0.199       |      0.128       |
>
> Regarding F1-score, it is calculated as $2 \cdot \frac{\text{precision} \cdot \text{recall}}{\text{precision} + \text{recall}}$, where precision and recall denote the percentage of points satisfying a distance threshold $\tau$, different from the Acc & Comp what we have evaluated. The specific value of the F1-score is significantly influenced by the threshold $\tau$. Since the 7-Scenes, NGRBD, DTU datasets do not provide official evaluation $\tau$ we temporarily decide not to select a threshold ourselves to avoid bias. If you deem an F1-score evaluation necessary for a fair comparison, we are willing to manually design reasonable thresholds $\tau$ for each dataset and provide the corresponding results.
>
> ---
> Unfinished. Please keep reading the comments

---

> ### Author Response · Authors · 2025-11-21
>
> ### **3. Clarification on CUT3R Baseline in Table 6**
> > CUT3R heavily relies on the sequential/temporal information. Thus, it is not a very good baseline here.
>
> We fully agree that CUT3R is explicitly designed for sequential data. We will update Table 6 in the final version.
>
> ### **4. Scale Alignment Strategy (Question)**
> > *...the other solution is to use umeyama to align the ground truth and predicted camera poses... Did the authors try this?*
>
> Yes, we explored this direction.
> 1. **Optimization Instability**: We initially attempted to supervise the model by applying Umeyama alignment between predicted and ground-truth camera poses. However, we found that this approach was difficult to converge, leading us to design the relative pose supervision which proved much more stable.
> 2. **Robustness of Scale Calculation**: Specifically regarding the calculation of the optimal scale $s^*$ (Eq. 4), using dense point maps ($N \times H \times W$ points) is significantly more robust than using sparse camera centers ($N$ points). Estimating scale from a large number of dense surface points minimizes the impact of outliers and noise compared to relying solely on a few camera positions.
>
> ---
> We appreciate your time and remain available to answer any further questions.

---

### Official Review · Reviewer_ghyt · 2025-11-01

**Soundness:** 4
**Presentation:** 4
**Contribution:** 4
**Rating:** 10
**Confidence:** 4

**Summary:**

This paper proposes a VGGT-like model that achieves superior performance across multiple tasks, including pose estimation, depth estimation, and point map estimation.

The performance improvement stems from a novel permutation-invariant model design. Previous methods such as Dust3R, Mast3R, and VGGT rely on a predefined reference frame during estimation, making them sensitive to frame selection and limiting their robustness. In contrast, this work attains permutation invariance by removing all frame order–dependent positional encodings and introducing a scale-invariant point map loss together with an affine-invariant pose loss.

Strengths: The proposed idea is both novel and insightful, supported by thorough and comprehensive experiments.

Weaknesses: No major issues are identified in this paper.

**Strengths:**

1. Novel and insightful idea: The paper makes an insightful observation about the limitation of defining a reference frame in prior models. The proposed approach to eliminate this dependency is technically sound and straightforward to implement.
2. Comprehensive experiments: The method is evaluated across multiple applications on widely used datasets. Reporting standard deviations in Table 6 is a valuable addition that strengthens the credibility of the results.
Well-presented paper: The paper is clearly written and easy to follow. Figures and tables are well designed, effectively supporting the main arguments and improving readability.

**Weaknesses:**

I don't see any major issues in this work.

**Questions:**

How is the rotation mapped to xyz in Figure 6? What's the rotation representation?

---

> ### Author Response · Authors · 2025-11-21
>
> We sincerely thank you for your strong endorsement and constructive feedback. It is truly encouraging to have the core contribution of our work—the permutation-equivariant design—recognized. We also appreciate your kind words regarding our experimental validation and the presentation of the paper.
>
> Response to Question:
>
> > *How is the rotation mapped to xyz in Figure 6? What's the rotation representation?*
>
> We apologize for the lack of clarity in the figure caption. We clarify the visualization method used in Figure 6 as follows:
> - Spatial Axes (XYZ = Translation): The spatial coordinates ($x, y, z$) of each point in the scatter plot represent the 3D translation (position) component of the camera pose, not the rotation.
> - Color (RGB = Rotation): The rotation component is independently visualized using RGB colors(4). specifically, we convert each predicted $3 \times 3$ rotation matrix into a 3D rotation vector (axis-angle representation). We then normalize the three components ($v_x, v_y, v_z$) of this vector to the range $[0, 1]$ across the dataset and map them to the Red, Green, and Blue channels, respectively.
> We will update the caption of Figure 6 in the final revision to include this detailed explanation.
>
> ---
> We appreciate your time and remain available to answer any further questions.

---

> > ### Comment · Reviewer_ghyt · 2025-11-26
> >
> > I appreciate the author's response and I have no further questions.

---

### Official Review · Reviewer_urpi · 2025-11-01

**Soundness:** 3
**Presentation:** 3
**Contribution:** 4
**Rating:** 6
**Confidence:** 5

**Summary:**

This paper proposes $\pi^3$, a feedforward model for pose estimation and point cloud prediction. It investigates the limitations of previous feedforward reconstruction models, such as VGGT, whose performance depends on the selection of the reference view. To address this issue, the proposed method removes the reference-view-dependent design in the model architecture and introduces a relative camera pose loss and a local point loss. Experimental results show that the proposed method effectively resolves the aforementioned problem and achieves state-of-the-art performance.

**Strengths:**

- The paper addresses an interesting and important problem: the dependence of previous feedforward reconstruction models on reference view selection. The proposed method provides a simple yet effective solution.

- The proposed model achieves SOTA performance on both pose estimation and reconstruction tasks.

**Weaknesses:**

- Some parts of the paper lack proper citations:
    - The normal loss is identical to that in MoGe, but no citation is provided.
    - The camera pose estimation module and loss function are the same as in Reloc3r, yet there is no citation in Section 3.3.

- The proposed method requires weight initialization from VGGT, which makes the comparison with other methods, such as VGGT itself, somewhat unfair. Table 8 shows results when both models are trained from scratch; however, it remains unclear how the model performs when trained from scratch only with the local point and pose losses. Furthermore, can the model trained from scratch with the added global loss achieve performance comparable to the version initialized from VGGT?

**Questions:**

- For monocular depth estimation, how does the performance compare with SOTA models such as DepthAnythingv2? Besides, what is the input image resolution used for depth estimation?
- Since the MLP head with pixel shuffling introduces grid-like artifacts, why does the proposed method still adopt this design instead of using DPT heads as in previous methods?

I will raise my score if the authors can provide reasonable clarifications regarding initialization and the effect of adding the global loss.

---

> ### Author Response · Authors · 2025-11-21
>
> We thank the reviewer for the positive assessment and the constructive feedback. Below, we address the specific concerns regarding initialization and citations.
>
> ###  1. **Response to Missing Citations**
> We sincerely thank the reviewer for pointing this out. This was indeed an oversight on our part. We will immediately add explicit citations and acknowledgments to MoGe (for the normal loss) and Reloc3r (for the camera pose module) in the final version of the manuscript to ensure academic rigor.
>
> ### 2. **Initialization Strategy and The Role of Global Loss (Crucial Clarification)**
> > *Performance of the model trained from scratch with local point loss and pose losses*
>
> As requested, we trained our model from scratch using only the affine-invariant camera pose and scale-invariant local point map losses (without the global point map regularization). For this experiment, we utilized the full dataset at a resolution of $224 \times 224$ and trained for 80 epochs (totaling 64,000 iterations). The results are compared below:
>
> | Method | ETH3D - Acc. ↓ | ETH3D Comp. ↓ | 7scenes Acc. ↓ | 7scenes Comp. ↓ | NRGBD Acc. ↓ | NRGBD Comp. ↓ |
> | :--- | :---: | :---: | :---: | :---: | :---: | :---: |
> | Ours (Pure Pi3 loss, Scratch, 80 epochs) | 0.618 | 0.453 | 0.064 | 0.068 | 0.071 | 0.047 |
> | VGGT (Scratch, 80 epochs) | 0.563 | 0.449 | 0.057 | 0.046 | 0.060 | 0.042 |
> | Ours (+ Global Proxy, Scratch, 80 epochs) | 0.418 | 0.266 | 0.059 | 0.071 | 0.052 | 0.035 |
>
> **Observation & Analysis**:
> We observe that the "Pure Pi3 loss," version performs worse than VGGT (scratch) and significantly worse than our version with the global auxiliary loss. We attribute this to the optimization difficulty inherent in relative pose supervision during the early training phase (the "cold start" problem).
> - **Coupling Issue**: Our method computes relative poses for all pairs, generating $N \times N$ supervision signals. While this design ensures permutation equivariance, optimizing these highly coupled constraints from a completely random initialization is unstable and slow to converge compared to methods anchored by a reference view.
> - **Alternative Validation (Warm Start)**: To verify that this is an optimization issue rather than a model defect, we performed a "warm start" experiment. We initialized our model using the weights from the "VGGT (Scratch)" experiment (trained for 80 epochs) and continued training with only Pi3 losses (trained for another 80 epochs). This yielded significantly better results (ETH3D Acc 0.437, 7-Scenes Acc 0.049, NRGBD Acc 0.053), confirming that the relative supervision struggles primarily with random initialization. However, simply training the "Ours (+ global proxy)" model for the equivalent duration (trained for 160 epochs) yields even better performance (ETH3D Acc 0.335, 7-Scenes Acc 0.041, NRGBD Acc 0.053). This comparison verifies that the global proxy is a more efficient strategy than multi-stage training for resolving the cold-start problem.
>
> ---
>
> Unfinished. Please keep reading the comments.

---

> ### Author Response · Authors · 2025-11-21
>
> > *Can the model trained from scratch with the added global loss achieve performance comparable to the version initialized from VGGT?*
>
> - **Role of Proxy Task**: The "Ours + Global Proxy" results in the table above demonstrate that introducing the global point map branch acts as a crucial proxy task for the scratch setting. It decouples the geometry learning to some extent, stabilizing the gradients and facilitating the optimization of the shared encoder, effectively solving the "cold start" problem. Notably, under this identical "from-scratch" setting, our model achieves performance that is competitive with, and on benchmarks like ETH3D, superior to the VGGT baseline (Acc 0.418 vs. 0.563). This indicates that our permutation-equivariant design remains highly effective compared to the reference-dependent baseline when the optimization landscape is stabilized.
> - **Why VGGT Initialization in Main Paper?** We chose to initialize from VGGT primarily to address the optimization challenges associated with the model scale. Our decoder accounts for a significant portion of the total parameters (~450M). Randomly initializing such a massive component drastically increases both the training difficulty and the computational cost. While introducing a global proxy (as demonstrated above) enables convergence and even outperforms the baseline, fully optimizing these randomly initialized parameters to reach their peak potential would still necessitate a substantially longer training duration. Therefore, we utilized the VGGT initialization as a practical strategy to accelerate convergence and reduce training overhead.
> - **Can the scratch model catch up?**
> Theoretically, yes, provided that the "from-scratch" model is given a data scale and training duration that exceeds the cumulative budget of the pre-trained baseline (i.e., VGGT's pre-training + our training).
>   Reason:
>   1. Architectural Superiority: As demonstrated in the table above, under the identical "from-scratch" setting, our model (with proxy) outperforms the VGGT baseline (e.g., ETH3D Acc 0.418 vs. 0.563). This confirms that our permutation-equivariant architecture is inherently more effective than the reference-dependent baseline when the cold-start issue is resolved.
>   2. Data & Compute Scaling: Since our model is superior under controlled, limited resources, it logically follows that if we were to train it with a data scale and duration comparable to VGGT's pre-training, it would likely match or even surpass the performance of the current VGGT-initialized version.
>   3. Current Reality: However, reproducing the massive pre-training phase of VGGT is practically infeasible, not only due to limited computational resources but also because VGGT was trained on large-scale internal datasets that are inaccessible to the public. Thus, utilizing pre-trained weights remains the only viable strategy to leverage those unavailable data priors and achieve SOTA performance efficiently.
>
> ---
> Unfinished. Please keep reading the comments.

---

> ### Author Response · Authors · 2025-11-21
>
> ### 3. **Monocular Depth Estimation**
> > *Comparison with SOTA models and Input Resolution*
>
> - Input Resolution: Following the CUT3R protocol, we resize input images to have a long edge of 518 pixels.
> - Comparison with Depth Anything V2: We compare $\pi^3$ with the SOTA monocular depth model, Depth Anything V2 (DAv2), on standard benchmarks.
>
> | Method             | Sintel Abs Rel↓ | Sintel delta < 1.25↑ | Bonn Abs Rel↓ | Bonn delta < 1.25↑ | KITTI Abs Rel↓ | KITTI delta < 1.25↑ | NYU-v2 Abs Rel↓ | NYU-v2 delta < 1.25↑ |
> | :----------------- | :---------------: | :--------------------: | :-------------: | :------------------: | :--------------: | :-------------------: | :---------------: | :--------------------: |
> | dav2metric_indoor  |       0.372       |         0.541          |      0.126      |        0.804         |      0.097       |         0.912         |       0.081       |         0.921          |
> | dav2metric_outdoor |       0.478       |         0.477          |      0.186      |        0.668         |      0.090       |         0.919         |       0.172       |         0.689          |
> | pi3                |       0.277       |         0.621          |      0.052      |        0.971         |      0.059       |         0.972         |       0.054       |         0.956          |
> | moge               |       0.273       |         0.693          |      0.050      |        0.976         |      0.054       |         0.977         |       0.055       |         0.952          |
> | mogev2             |       0.281       |         0.675          |      0.068      |        0.969         |      0.052       |         0.979         |       0.065       |         0.932          |
>
> Analysis: As shown, $\pi^3$ achieves comparable performance to the specialized DAv2, despite being designed for generalist multi-view reconstruction.
>
> ### 4. **MLP Head vs. DPT Head**
> > *Reason for adopting MLP head despite grid-like artifacts*
>
> The primary motivation for this design is efficiency. DPT or convolutional heads (e.g., as used in MoGe) are computationally heavy and memory-intensive. The "MLP + pixel shuffling" design significantly reduces VRAM usage, which ensures fast training and inference. This efficiency is particularly important as it allows for scalable multi-task extensions (such as the global proxy branch discussed earlier) without memory overflow. While we acknowledge the grid-like artifacts, we believe that developing better upsampling methods for transformer-based dense prediction remains a valuable open research direction.
>
> ---
> We appreciate your time and remain available to answer any further questions.

---

### Author Response · Authors · 2025-12-03
**Summary**

Dear Reviewers and AC,

We sincerely thank the Area Chair and the reviewers for their constructive feedback and time. We have comprehensively addressed all concerns raised by the reviewers, ranging from experimental validation and baseline comparisons to methodology clarification. These revisions have significantly strengthened the rigor and completeness of our work.

Below is a summary of the key revisions we have made in the final version:
- Citations and Acknowledgments (Section 3): We have added explicit citations to MoGe and Reloc3r in the methodology section to properly acknowledge their contributions to the training loss design, as pointed out by Reviewer urpi.
- Additional Baselines and Analysis (Appendix A.4): As suggested by Reviewer urpi, we have updated Table 8 to include the results of the model trained from scratch using only affine-invariant camera pose and scale-invariant local point map losses. We have also added a detailed analysis of the "global proxy" strategy and its critical role in resolving the optimization cold-start problem.
- Improved Visualization Details (Appendix A.3): Addressing Reviewer ghyt's comment, we have expanded Appendix A.3 to provide a detailed explanation of the methodology used for visualizing the 3D camera pose distributions (spatial coordinates for translation and RGB mapping for rotation).
- Additional Evaluation Results (Appendix A.7): We have extended our original evaluation results. Per the suggestion of Reviewer urpi, we add the monocular depth evaluation of Depth Anything V2 under our standard benchmark, adhering to the CUT3R evaluation protocol. In response to Reviewer DanB's comment, we add Chamfer Distance evaluation for all methods on all datasets we used in point map estimation; we also evaluate camera pose under tighter angular thresholds (1°, 3°, 5°, 10°, 15°), which provides a more granular and comprehensive view of precision.
- Fair Comparison (Table 6): We have removed the CUT3R results from Table 6 to ensure a strictly fair comparison with other baselines, as recommended by Reviewer DanB.

We believe these updates resolve all pending issues and ensure the paper meets the high standards of the conference. We hope the AC finds the revised manuscript suitable for acceptance.

Best Regards,
Authors.

---

### Meta-Review · Area_Chair_4Dq7 · 2025-12-28

**Summary:**

This paper introduces a feed-forward neural network for visual geometry reconstruction. Most reviewers are positive on this work but it still invovles some concerns, like method clarity, unfair comparisons etc. Authors should ensure the proper revision of the paper.

Some specific aspects include:
The papers requires weight initialization from VGGT, which makes the comparison with other methods, such as VGGT itself, somewhat unfair. Try to use umeyama to align the ground truth and predicted camera poses.

**Reviewer Scores:**

NA

---

### Decision · Program_Chairs · 2026-01-26

Accept (Poster)